# Exploring mental health symptoms in elite athletes during the COVID-19 pandemic: A systematic review and meta-analysis on sex differences

Liang-Tseng Kuo [1,2‡], Sung-Huang Laurent Tsai[2,3‡], Udit Dave[4], William A. Marmor[5], Reena Olsen[6], Bridget Jivanelli[7], Michelle E. Kew[6], Daphne I. Ling[6,8,9]*

1 Department of Sports Medicine, Landseed International Hospital, Taoyuan, Taiwan, 2 School of Medicine, Chang Gung University, Taoyuan, Taiwan, 3 Department of Orthopaedic Surgery, Chang Gung Memorial Hospital, Keelung Branch, Keelung, Taiwan, 4 Tulane University School of Medicine, New Orleans, Louisiana, United States of America, 5 Department of Orthopaedics, University of Miami, Miami, Florida, United States of America, 6 Sports Medicine Institute, Hospital for Special Surgery, New York, New York, United States of America, 7 Kim Barrett Memorial Library, HSS Education Institute, Hospital for Special Surgery, New York, New York, United States of America, 8 Department of Public Health, College of Medicine, National Cheng Kung University, Tainan, Taiwan, 9 Department of Population Health Sciences, Weill Cornell Medical College, New York, New York, United States of America

‡ LTK and SHLT contributed equally to this work as co-first authors.
* wosohealth@gmail.com

**Data Availability Statement:** All relevant data are within the paper and its Supporting Information files.

## Abstract

The COVID-19 pandemic significantly affected elite athletes, leading to increased mental health issues such as stress, anxiety, and depression. Sex differences in mental health may exist among athletes during the COVID-19 crisis. This study aimed to perform a systematic review and meta-analysis to examine sex differences in mental health symptoms among elite athletes during the COVID-19 pandemic. We systematically searched the databases including Pubmed, EMBASE, and manually checked previous systematic reviews for relevant studies in March 2024. Authors were also contacted for sex-specific data. Studies were included if they compared mental health symptoms between male and female elite athletes during the COVID-19 pandemic. We used a random-effects model to summarize the rate ratio (RR) between female and male athletes across studies. Risk of bias in studies was assessed using a 9-item tool. We included 18 studies in this review. The results indicated that female athletes reported higher levels of anxiety (RR 1.24, 95% CI = 1.08 to 1.43) and depression (RR 1.36, 95% CI = 1.15 to 1.61) than male athletes during the pandemic. They also had a higher risk of stress or distress (RR 1.27, 95% CI = 0.99 to 1.63) than their male counterparts. No significant differences were found regarding alcohol use or misuse between female and male athletes (RR 1.01, 95% CI = 0.75 to 1.37). Limited evidence was available for eating disorders, gambling, substance use, and sleep problems. In conclusion, female and male athletes differed in mental health symptoms during the COVID-19 pandemic. Female athletes were more likely to report anxiety, depression, and distress. Incorporating mental health resources may be particularly

**Funding:** The author(s) received no specific funding for this work.

**Competing interests:** he authors have declared that no competing interests exist.

important for women's sports, which have smaller financial margins and greater career uncertainty than men's sports.

## Introduction

The COVID-19 pandemic led to a significant increase in mental health problems globally [1–4]. The widespread physical isolation, financial uncertainties, and continuous threat of infection contributed to a variety of mental health issues such as anxiety, depression, stress, feelings of isolation, and substance abuse [5–7]. The pandemic's impact on mental health varied among different populations, with specific groups experiencing greater vulnerability to mental health problems, such as frontline healthcare workers, the elderly, and those with pre-existing mental health conditions [8].

Elite athletes are a unique subgroup that was particularly affected by the pandemic, as the cancellation of organized sporting events, disruptions in training routines, and a lack of social support contributed to an increase in mental health issues within this population [9–11]. Athletes faced unique challenges, including loss of identity, purpose, and motivation due to the sudden halt in their careers, leading to heightened stress, anxiety, and depression [12, 13]. Additionally, the uncertainty surrounding the resumption of events and the potential impact on their future careers compounded these mental health problems. Moreover, there may be a disproportionate impact of the COVID-19 pandemic on women's sports compared to men's sports. There are significant disparities in sponsorship income, financial resources, professional career opportunities, and global media coverage between women's and men's sports [14, 15]. These pre-existing differences are likely to result in female athletes facing even more stress during the COVID-19 crisis than their male counterparts.

The COVID-19 pandemic has highlighted the need to understand the unique mental health challenges faced by athletes during extreme situations, particularly regarding the sex-based disparities in mental health symptoms. This systematic review and meta-analysis aims to explore these disparities and provide insights to inform future interventions and support strategies for athletes during crises like the COVID-19 pandemic.

## Methods

We conducted this systematic review and meta-analysis according to the Cochrane recommendation [16] and reported it per the Preferred Reporting Items for Systematic Reviews and Meta-Analyses (PRISMA) 2020 guideline [17].

### Literature search

The initial search strategy was developed by a medical librarian (BJ) using MEDLINE (PubMed) and was subsequently adapted for other databases, including Embase and the Cochrane Library (which includes the Cochrane Central Register of Controlled Trials, Health Technology Assessment database, and NHS Economic Evaluation Database).

The search terms used in this study were a combination of keywords, including COVID-19, SARS-CoV-2, Severe Acute Respiratory Syndrome Coronavirus 2, coronavirus, Sex, male, men, female, woman, college, professional, elite, athlete, sport, mental health, sleep disturbance, anxiety, depression, and alcohol/drug/substance use, misuse, or abuse. The complete search strategy can be found in **S1 Table**.

This systematic review included studies published between January 2010 and March 2024 that provided data on mental health symptoms (including alcohol use, anxiety, depression,

distress, eating disorders, gambling, sleep problems, and substance use) among elite athletes in head-to-head comparisons between male and female athletes. The eight symptoms were based on a recent consensus statement from the International Olympic Committee on mental health in elite athletes [18]. The chosen search strategy was based on a previous meta-analysis of mental health in athletes (manuscript under review) that included studies published from January 2010 to October 2021. The search was then updated to March 2024 to include COVID-19 studies. Conference abstracts and studies that did not present sex-based comparisons, focused on non-elite athletes (such as youths or adolescents), or not written in English were excluded. Two authors (LTK and SHLT) independently screened titles, abstracts, and full-article texts using the online software program Covidence (Veritas Health Innovation Ltd; Melbourne, Australia). In addition, two recent systematic reviews were manually searched for additional studies [19, 20]. Any disagreements were resolved through consensus with the senior author (DL). The co-authors (LTK, SHLT, DL) extracted the following data from full-text articles: study design, number of athletes, team or individual sport, and mental health symptoms. The co-authors also contacted the corresponding authors for articles that were relevant but did not provide sex-specific data.

**Data synthesis and meta-analysis.** For each study, two pairs of values were extracted: the total of male and female athletes, and the number of male and female athletes exhibiting each specific mental health symptom. Rate ratios (RRs) were calculated by dividing the rate in female athletes by the rate in male athletes. Thus, RRs greater than 1 indicated a higher symptom burden for women, while RRs below 1 implied a greater symptom burden for men. The DerSimonian-Laird random-effects model (REM) was used to pool data when there were more than 3 studies for a mental health symptom. This approach offers more conservative estimates with wider confidence intervals compared to fixed-effects models [21], due to the REM accounting for both inter- and intra-study variability. The raw data for the studies included in this meta-analysis are available upon request.

**Risk of bias assessment.** A 9-item tool was used to evaluate the risk of bias in the included studies [22]. Each potential bias was rated as low (0) or high (1). The scores were then summed to determine the quality of the study. The total score was categorized as follows: low risk of bias with a score of 0–3, moderate risk 4–6, and high risk 7–9. Two authors (LTK and SHLT) assessed the risk of bias, and any disagreements were resolved through consensus.

## Results

### Included studies

The process of study selection is presented in **Fig 1**. The initial search returned 2148 articles, and eight additional articles were identified from the reference lists of previous systematic reviews [19, 20]. After removing 749 duplicates, 1407 articles were screened by titles and abstracts. Among the 64 articles that remained, 46 were excluded and 18 studies were found to meet our inclusion criteria. A list of the 18 included studies is shown in Table 1.

### Characteristics of included studies

We included 18 studies [9, 13, 23–38] published between 2020 to 2022. Two studies enrolled athletes from individual sports only [9, 25], two studies enrolled athletes from team sports only [13, 30], and the remaining studies evaluated both team and individual sports. In terms of level of competition, two studies included professional athletes [9, 13], four included college athletes [23, 28, 32, 33], and the remaining included national team athletes.

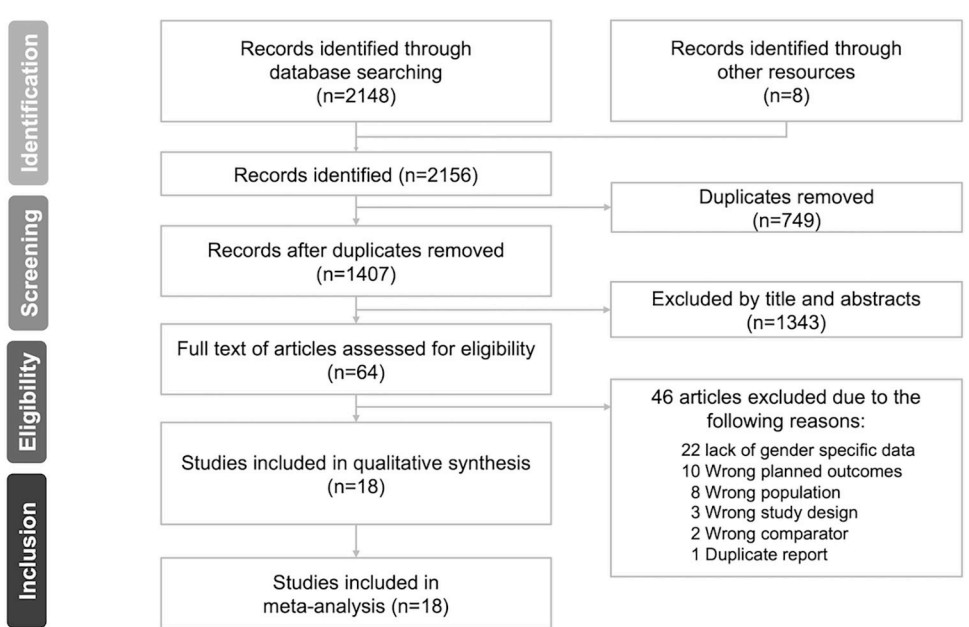

**Fig 1. PRISMA (Preferred Reporting Items for Systematic Reviews and Meta-Analyses) flow diagram of the study.**

## Outcomes

**Alcohol misuse.** Five studies with 7014 participants (4492 females and 2522 males) reported alcohol use or misuse in elite athletes (**Fig 2**) [13, 23, 28, 29, 32]. The results showed no significant differences between male and female athletes regarding alcohol use/misuse (RR 1.01, CI = 0.75 to 1.37). Imboden et al. followed a cohort of 193 national team athletes in 2020, with male athletes reporting a higher risk for alcohol use (RR 0.69, CI = 0.52 to 0.91) during the COVID-19 lockdown than female athletes [29].

**Anxiety.** A total of nine studies with 3729 participants (1462 females and 2257 males) were included (**Fig 3**) [9, 13, 23, 25, 31, 34, 35, 37, 38]. The results showed that female athletes had a higher overall risk of anxiety compared to male athletes (RR 1.24, CI = 1.08 to 1.43). Hakansson et al. enrolled professional athletes and found that female athletes had a significant risk of reporting anxiety (RR 3.80, CI = 1.80 to 8.01) [13].

**Depression.** Depression was assessed in ten studies (**Fig 4**) [9, 13, 23, 25, 30, 31, 34–37]. The results showed that female athletes had a significantly increased risk of depression (RR = 1.35, 95%CI = 1.14 to 1.58) compared to male athletes. Håkansson et al. reported data from professional athletes and similarly found that female athletes had greater depression than male athletes during the COVID-19 pandemic (RR 5.15 CI = 2.12 to 12.53) [13].

**Distress/stress.** Distress or stress was evaluated in six studies [23–27, 34]. Female athletes showed a trend of higher stress or distress levels (RR 1.27, CI = 0.99 to 1.63) than male athletes during COVID-19 (**Fig 5**). Parm Ü et al. reported on 102 elite athletes in both team and individual sports and showed that women had more distress symptoms compared to men (RR 3.92, CI = 1.79 to 8.56) [34]. In the only study to report data for athletes who had tested positive for COVID-19, Buonsenso et al. also showed that female athletes had higher risk of distress (RR 1.30, CI = 1.15 to 1.47) compared to male athletes [24].

**Other mental health symptoms.** Substance use was reported in two studies with a total of 392 participants (184 females and 198 males) [29, 32]. There were low rates of drug abuse in the athlete population. Imboden et al. found no significant difference in marijuana use

**Table 1. Characteristics of included studies.**

| First Author, Year | Study Design | Country | Sport | Level | Mean age, year (range) | Number of Athletes (F/M) | Current or former | Mental health symptom | Method of measurement |
|---|---|---|---|---|---|---|---|---|---|
| Boudreault 2022 [23] | Cross-sectional | Canada | Team & Individual | College | 21.8 ± 2.6 (18–43) | 423 (266/157) | Current | Alcohol misuse Anxiety Depression Eating Disorder Stress | AUDIT-C GAD-7 PHQ-9 EAT-26 PSS-10 |
| Buonsenso 2022 [24] | Retrospective cohort | Italy | Team & Individual | National | 24.96 ± 9.82 | 204 (93/111) | Current | Stress | GHQ-12 |
| Davila-Torres 2021 [25] | Cross-sectional | Peru | Rugby | National | 20.5 ± 4.4 | 74 (32/42) | Current | Anxiety Depression Stress | DASS-21 |
| Di Cagno 2020 [26] | Cross-sectional | Italy | Team & Individual | National | 27.59 ± 10.73 | 368 (155/213) | Current | Stress | IES-R |
| Fiorilli 2021 [27] | Cross-sectional | Italy | Team & Individual | National | >18 | 263 (127/136) | Current | Stress | IES-R |
| Gouttebarge 2022 [9] | Cross-sectional | Netherlands | Soccer | Professional | F: 22.8 ± 3.9, M: 26.0 ± 5.0 | 1602 (468/1134) | Current | Depression, Anxiety | GAD-7, PHQ-9 |
| Guillot 2022 [28] | Cross-sectional | United States | Team & Individual | College | >18 | 5899 (3924/1975) | Current | Alcohol misuse | AUDIT-C |
| Hakansson, 2020 [13] | Cross-sectional | Sweden | Team | Professional | >15 | 310 (118/192) | Current | Alcohol use, Anxiety Depression Concern about the future | PHQ-9, GAD-7, questionnaire |
| Imboden 2021 [29] | Cross-sectional | Swiss | Team & Individual | National | 24.1 ± 5.2 | 193 (87/106) | Current | Alcohol use, Alcohol use, increase Drug, marijuana use | Custom survey |
| Ivarsson 2021 [30] | Prospective cohort | Italy | Football | National | 22.4 ± 5.2 | 47 (26/21) | Current | Depression Distress | WHO-5 PANAS |
| Knowles 2021 [31] | Cross-sectional | Ireland & United Kingdom | Team & Individual | National | >18 | 360 (161/199) | Current | Anxiety, Depression | HADS |
| McLellan 2022 [32] | Cross-sectional | United States | Team & Individual | College | NA | 189 (97/92) | Current | 1.Alcohol misuse 2.Drug use, (marijuana, other) 3. Sleep, not restful/adequate | 1–2. self-report 3.custom question, strongly disagree |
| Melone 2022 [33] | Cross-sectional | France | Team & Individual | College | 20 (IQR 19–21) | 338 (164/174) | Current | Sleep quality | SDS, PSQI |
| Parm Ü 2021 [34] | Cross-sectional | Estonia | Team & Individual | National | 24.7 ± 8.6 (16–60) | 102 (58/44) | Current | Anxiety Depression, Insomnia | EST-Q2 |
| Pensgaard 2021 [35] | Cross-sectional | Norway | Team & Individual | National | 26.9 (18–59) | 378 (159/219) | Current | Anxiety, Depression, Insomnia, Eating disorder, Gambling | HSCL-1, BIS, EDE-QS, CPGI |
| Pillay 2020 [36] | Cross-sectional | South Africa | Team & individual | National | >18 | 698 (225/463) | Current | Depression | "Do you feel depressed" Yes/no |
| Salles 2022 [37] | Cross-sectional | United Kingdom | Team & Individual | National | 25.6 ± 4.9 (18–27) | 274 (110/164) | Current | Anxiety, Depression, Sleep | GAD-7 PHQ-9 ISI-7 |

(*Continued*)

**Table 1.** (Continued)

| First Author, Year | Study Design | Country | Sport | Level | Mean age, year (range) | Number of Athletes (F/M) | Current or former | Mental health symptom | Method of measurement |
|---|---|---|---|---|---|---|---|---|---|
| Soares 2021 [38] | Cross-sectional | Brazil | Team & Individual | National | 23 (18–57) | 206 (90/116) | Current | Anxiety state Anxiety trait, Sleep quality | STAI self-report |

Abbreviations

AUDIT-C Alcohol Use Disorders Identification Test-Concise, BIS Bergen Insomnia Scale, CPGI Canadian Problem Gambling Index, DASS-21 Depression, Anxiety, and Stress scale-21, EAT-26 Eating Attitudes Test–26, EDE-QS Eating Disorder Examination Questionnaire Short, EST-Q2 Emotional state questionnaire, GAD-7 Generalized Anxiety Disorder-7-Item, HADS Hospital Anxiety and Depression Scale, HSCL-1 Hopkins Symptoms Check List – 10, ISI-7 Insomnia Severity Index, NA not available, PHQ-9 Patient Health Questionnaire-9, PSS-10 Perceived Stress Scale-10, PSQI Pittsburg Sleep Quality Index, SDD Sleep Difficulty Score, STAI State-Trait Anxiety Inventory.

between female and male athletes (RR 0.17, CI = 0.009 to 3.32) [29], and McLellan et al. also reported similar findings for marijuana and other drugs [32]. Two studies examined athletes for eating disorders [23, 35]. Pensgaard et al. found that female athletes had a significantly higher risk of eating disorders (RR 6.43, CI = 1.88 to 21.99) [35]. Boudreault et al. showed a similar sex-difference, however it was not significant (RR 2.07, CI = 0.85 to 5.01) [23].

Gambling was also reported in two studies [13, 35]. Håkansson et al. did not find a difference in gambling problems between female and male professional athletes (RR 0.41, CI = 0.14 to 1.19) [13], whereas Pensgaard et al. reported that female athletes had a lower risk of gambling (RR 0.14, CI = 0.03 to 0.61) [35]. Sleep problems were evaluated in seven studies [32–38]. Three studies reported insomnia [34, 35, 37], two studies reported inadequate or unrestful sleep [32, 36], and the remaining two studies reported poor sleep quality [33, 38]. The results are presented in **Fig 6** by subgroup. While the evidence shows that female athletes tend to have a higher risk of sleep problems, including insomnia [34] and poor quality sleep [33], the limited number of studies prevents reporting of conclusive findings.

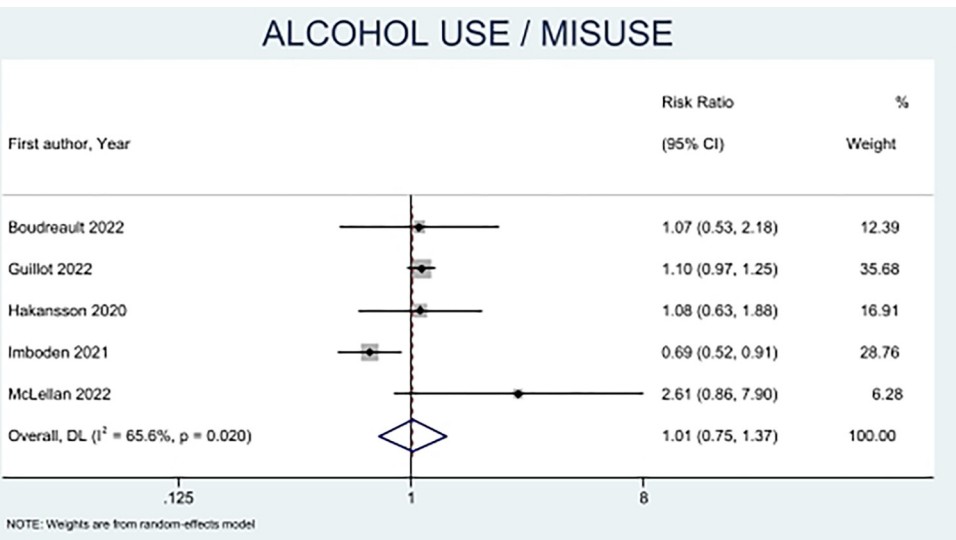

**Fig 2. Forest plot of sex-based differences for alcohol use or misuse.**

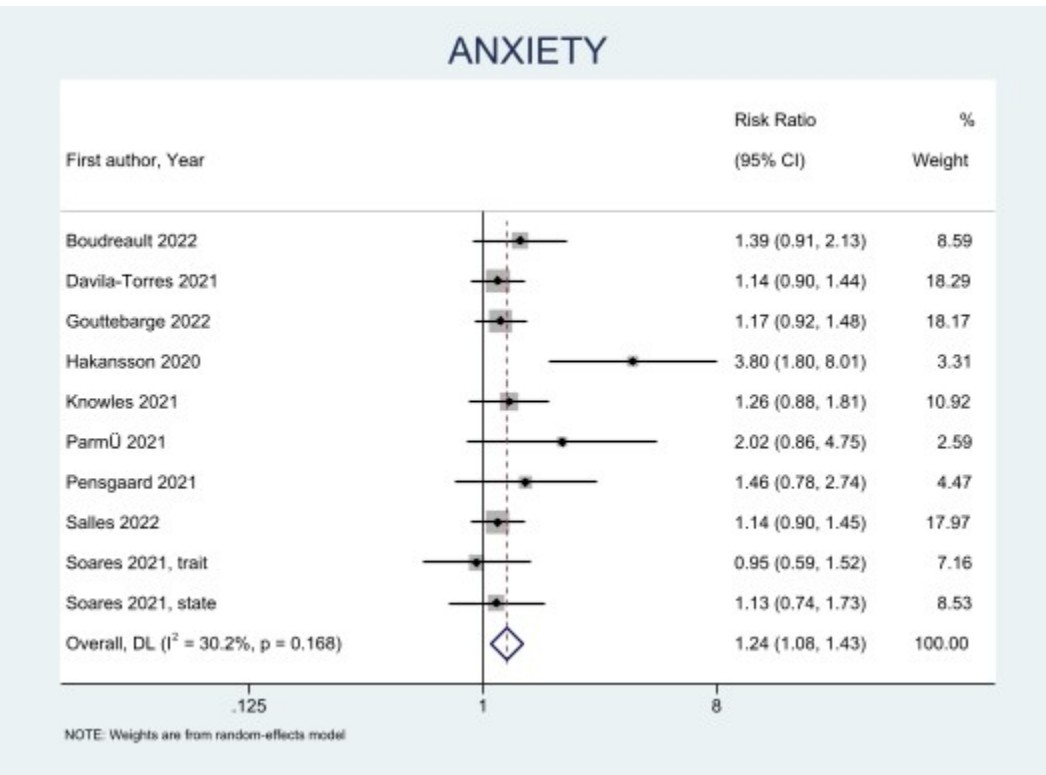

**Fig 3. Forest plot for sex-based differences in anxiety.**

Each of the 18 original studies included in this systematic review has a low overall risk of bias (**S2 Table**).

## Discussion

This study compared mental health symptoms between elite male and female athletes during the COVID-19 pandemic. The meta-analysis showed that female athletes have an overall increased risk of reporting anxiety, depression, and distress. There was no significant difference in alcohol use between male and female athletes. Additionally, there was limited evidence for the sex-based differences in other mental health symptoms that we evaluated: eating disorders, gambling, substance use, and sleep problems.

In this meta-analysis, depression and anxiety were the mental health conditions that were most frequently evaluated. Female athletes experienced depression and anxiety symptoms at a rate that was 1.36 and 1.24 times higher, respectively, than male athletes. Biological differences, societal expectations, and gender roles may interact with pandemic-related stressors, exacerbating the risk of greater depression and anxiety [39, 40]. In addition, depressive symptoms were found to be associated with worse sports performance and difficulty coping with restrictions as well as more symptoms of anxiety, sleeping issues, and financial fears [41].

During the COVID-19 pandemic, female athletes also tended to report higher levels of stress or distress than male athletes, which has been associated with an overall rise in anxiety and depression rates [42]. The inherent pressures of elite sports and uncertainties surrounding athletes' careers due to abrupt changes during the pandemic may have contributed to heightened psychological distress in female athletes [9], which were only exacerbated by other factors such as disruptions to training routines, cancellations or postponements of competitions,

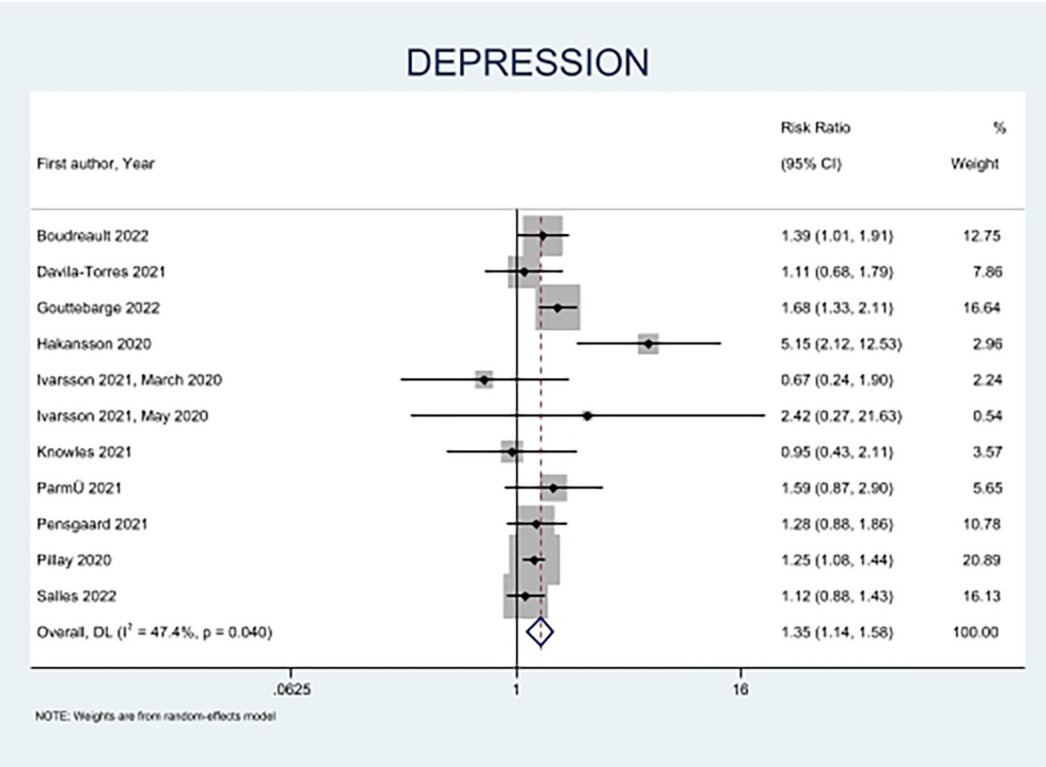

**Fig 4. Forest plot of sex-based differences for depression.**

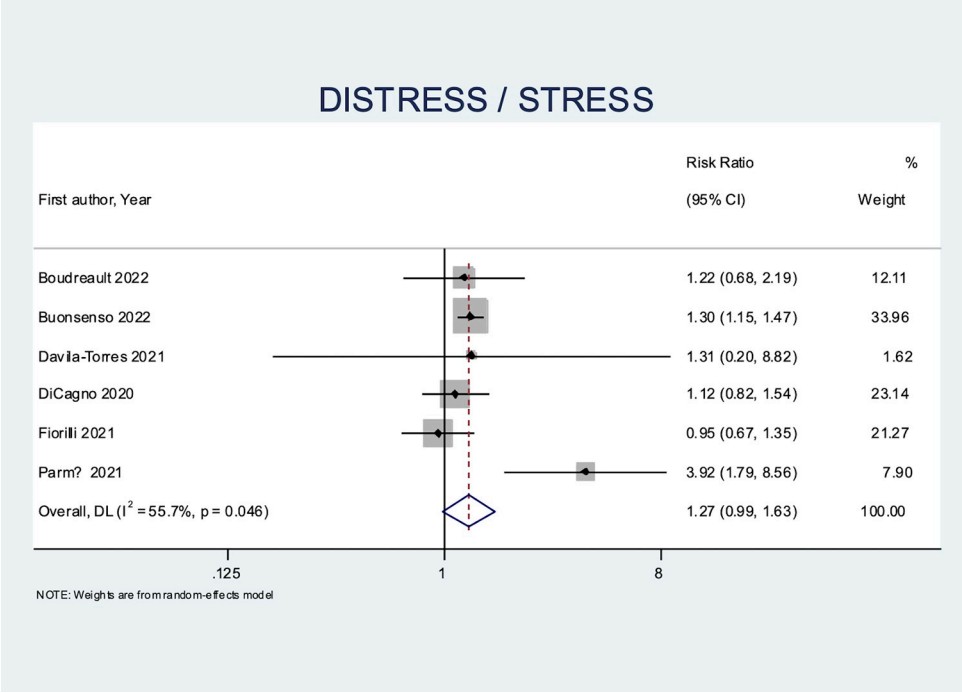

**Fig 5. Forest plot of sex-based differences for distress/stress.**

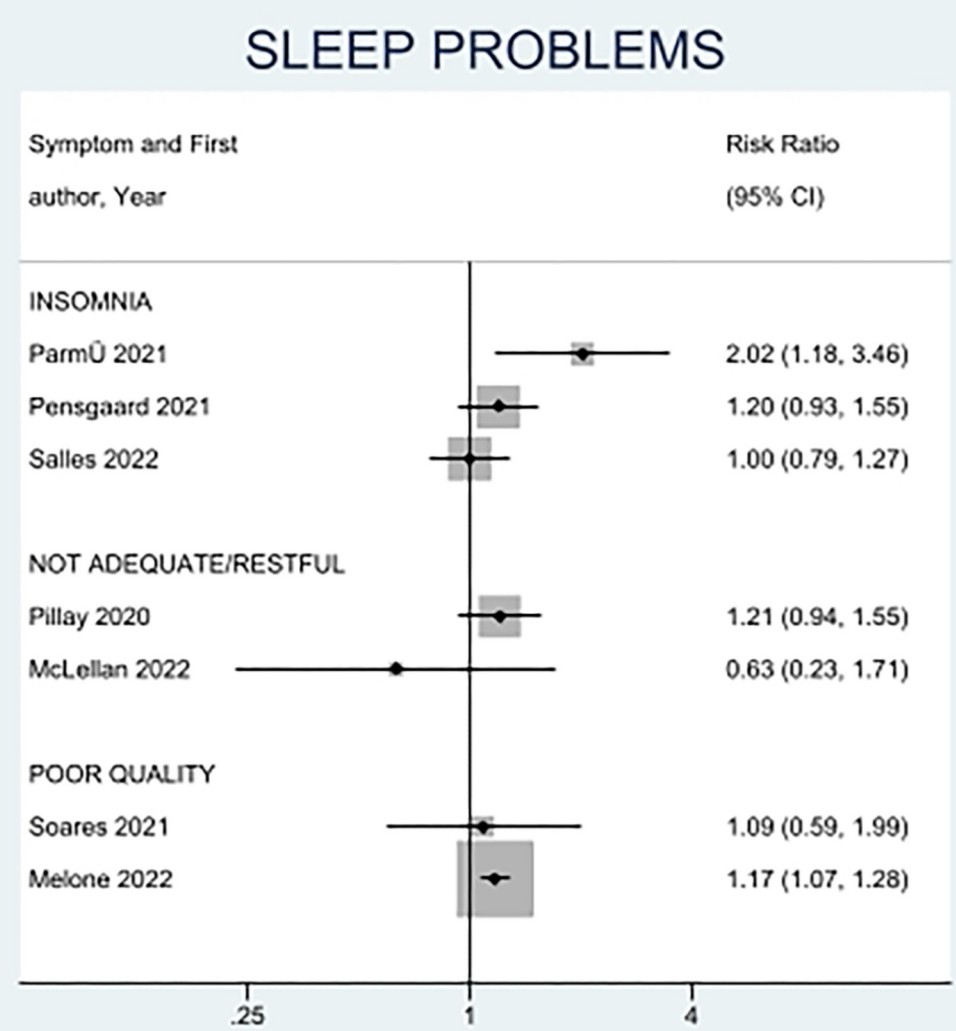

**Fig 6. Forest plot of sex-based differences in sleep problems.**

insufficient social support, financial worries, and health concerns [36, 43–47]. Acknowledging and addressing these factors is crucial for supporting elite athletes' mental health and well-being during unique situations such as the COVID-19 pandemic.

College and professional athletes frequently experience sleep difficulties, with more than 80% of them experiencing daytime tiredness and poor sleep quality [48]. There is limited evidence suggesting that female athletes have greater risk factors for insomnia and poor sleep quality. This disparity may be attributed to biological factors, such as hormonal fluctuations and the increased prevalence of depression and anxiety among female athletes [49–51], as mentioned previously. Pandemic-related disruptions in daily routines, increased stress, and reduced physical activity may further exacerbate sleep disturbances in athletes [52, 53].

No differences were found in alcohol misuse between elite male and female athletes in the meta-analysis. Only Imboden et al. showed that male sex is a well-known risk factor for increased alcohol use in the general population [29]. Furthermore, a meta-analysis presently under review that evaluated mental health symptoms in athletes outside of the COVID-19 period included 9 studies which showed that male athletes had greater risk of alcohol use (RR

0.83, 95% CI = 0.74 to 0.94). There was also evidence of increased alcohol binging and intoxication in male athletes (data not shown). This finding suggests that the pandemic may have caused a change in behavior among female athletes, in this case increased alcohol use compared to pre-pandemic levels.

Our investigation highlights the need for taking a gendered approach to adverse mental health outcomes in elite athletes. This gender discrepancy was only exacerbated by the pandemic's restrictive measures, including lockdowns, disrupted training regimens, and cancelled competitions, all of which have disproportionately affected female athletes' mental health [51, 54, 55]. The many challenges of women's sports arise from inherent disparities in sponsorship income, financial resources, professional career prospects, and international media coverage when compared to men's sports [14, 15]. In the only longitudinal study that was included in this meta-analysis, Ivarsson et al. found that male athletes had greater depression in the early stage of lockdown, but this trend reversed many weeks later [30]. While the RRs were not significant, this finding may suggest that depressive symptoms persisted more in female athletes as restrictions gradually lifted and training returned to a new normal. The implementation of mental health initiatives and provision of resources to aid elite athletes, with a particular focus on female athletes, can mitigate the psychological symptoms that they normally encounter during their careers and may help alleviate the negative impact of substantially abnormal events such as a pandemic lockdown [47].

## Strengths and limitations

This study has numerous strengths. Using the International Olympic Committee's recent development of two mental health screening tools, we evaluated eight mental health symptoms in this work. To ensure that all relevant studies were considered, an experienced medical librarian (BJ) searched multiple databases, complemented by a manual review of relevant systematic reviews for further references and contacting authors to capture the most data from this recent historic event. Moreover, we extracted data that enabled direct comparisons between male and female athletes across a broad spectrum of mental health symptoms. Consequently, our comparative findings are not based on aggregate data derived from disparate populations or settings.

There were also several limitations in our study. First, a major limitation was the variability of scales used by the included studies to assess each mental health symptom, which could contribute to the substantial heterogeneity observed in our results. Furthermore, certain mental health symptoms featured a limited number of studies, thereby prohibiting further meta-analysis and subgroup analysis to investigate potential sources of heterogeneity, such as geographic region or sport type. Second, our study was limited by data unavailability for specific comparisons. Stratifying our analysis based on team versus individual sports was not possible, as most studies incorporated both types. Previous research has indicated that individual sports have distinct impacts on mental health compared to team sports [51, 56–58]. In fact, female athletes participating in individual sports exhibited a higher risk for depression [56, 57]. Similarly, Schaal et al. discovered a higher rate of anxiety and depression among athletes in individual sports [51].

An additional limitation pertains to the cross-sectional design of nearly all of the included studies. We could not evaluate the progression of these mental health symptoms at various stages of the pandemic lockdown, nor could we determine if gender differences in these mental health symptoms persisted as the pandemic progressed. A meta-analysis of longitudinal cohort studies in the non-athletic population comparing mental health before and during the pandemic revealed a slight increase in mental health symptoms shortly after the COVID-19

pandemic began, which then decreased and became comparable to pre-pandemic levels by mid-2020 among most populations [59]. This trend may be attributed to effective resilience and adaptation; however, significant heterogeneity existed among subgroups, and time-lag effects may also be present [60]. Similarly, we were not able to stratify our results by infection status, as nearly all of the studies did not include this type of analysis. A final limitation is that we evaluated mental health symptoms in isolation when in reality, they may be interconnected and exacerbate each other clinically.

## Conclusion

Female and male athletes have significant differences in reported mental health symptoms during the COVID-19 pandemic. Female athletes are more likely to report anxiety, depression, and distress in comparison to male athletes. Alcohol use was generally similar. It is crucial to address this sex-based disparity by providing tailored interventions and support strategies to mitigate the negative impact on female athletes' mental health and performance during major disruptive events such as a pandemic. Diligent observation and assessment of athletes' mental health is a vital component of any sport and should incorporate mental health resources such as mindfulness-based initiatives, stress management, or resilience assistance [18].

## Supporting information

**S1 Checklist. PRISMA 2020 checklist.**
(DOCX)

**S1 Table. Database search strategy.**
(DOCX)

**S2 Table. Risk of bias assessment.**
(DOCX)

**S1 Data.**
(CSV)

**S2 Data.**
(CSV)

## Author Contributions

**Data curation:** Liang-Tseng Kuo, Sung-Huang Laurent Tsai, Udit Dave, William A. Marmor, Reena Olsen, Bridget Jivanelli.

**Formal analysis:** Daphne I. Ling.

**Methodology:** Liang-Tseng Kuo, Bridget Jivanelli, Daphne I. Ling.

**Supervision:** Michelle E. Kew, Daphne I. Ling.

**Writing – original draft:** Liang-Tseng Kuo, Sung-Huang Laurent Tsai.

**Writing – review & editing:** Michelle E. Kew, Daphne I. Ling.

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
