## [Decision Letter · Decision Letter 0]

24 Sep 2024

PONE-D-24-28145Exploring Mental Health Symptoms in Elite Athletes During the COVID-19 Pandemic: A Systematic Review and Meta-Analysis on Sex DifferencesPLOS ONE

Dear Dr. Ling,

Thank you for submitting your manuscript to PLOS ONE. After careful consideration, we feel that it has merit but does not fully meet PLOS ONE’s publication criteria as it currently stands. Therefore, we invite you to submit a revised version of the manuscript that addresses the points raised during the review process.

We look forward to receiving your revised manuscript.

Comments from PLOS Editorial Office: We note that one or more reviewers has recommended that you cite specific previously published works. As always, we recommend that you please review and evaluate the requested works to determine whether they are relevant and should be cited. It is not a requirement to cite these works. We appreciate your attention to this request.

Kind regards,

Efrem Kentiba, PhD

Academic Editor

PLOS ONE

2. As required by our policy on Data Availability, please ensure your manuscript or supplementary information includes the following:

Additional Editor Comments (if provided):

Reviewers' comments:

Reviewer's Responses to Questions

**Comments to the Author**

1. Is the manuscript technically sound, and do the data support the conclusions?

Reviewer #1: Yes

Reviewer #2: Yes

Reviewer #3: Yes

2. Has the statistical analysis been performed appropriately and rigorously? 

Reviewer #1: Yes

Reviewer #2: Yes

Reviewer #3: Yes

3. Have the authors made all data underlying the findings in their manuscript fully available?

Reviewer #1: Yes

Reviewer #2: No

Reviewer #3: Yes

4. Is the manuscript presented in an intelligible fashion and written in standard English?

Reviewer #1: Yes

Reviewer #2: Yes

Reviewer #3: Yes

5. Review Comments to the Author

Reviewer #1: A competent and clinically relevant systematic review and meta-analysis on sex differences in mental health symptoms in elite athletes during the COVID-19 Pandemic. This will be of specific interest to sports medicine.

Reviewer #2: dear researchers

please consider the following points:

1- check key words in the abstract based on the Mesh standard.

2- update the relevant reference about mental health of athletes during the pandemic. the following reference sare recommended:

Rassolnia, Atie, and Hadi Nobari. "The impact of socio-economic status and physical activity on psychological well-being and sleep quality among college students during the COVID-19 pandemic." International Journal of Sport Studies for Health 7.2 (2024): 1-12.

Taheri, Morteza, et al. "Mental health, eating habits and physical activity levels of elite Iranian athletes during the COVID-19 pandemic." Science & Sports 38.5-6 (2023): 527-533.

Taheri, Morteza, et al. "Comparative study of the long-term impact of the COVID-19 pandemic on mental health and nutritional practices among international elite and sub-elite athletes: a sample of 1420 participants from 14 countries." Sports medicine-open 9.1 (2023): 104.

3- The reliance on a wide range of assessment tools (such as the GAD-7, PHQ-9, etc.) adds heterogeneity to the results, which could limit the comparability of findings across studies. While the authors acknowledge this, a more uniform approach or further subgroup analysis might have strengthened the results.

4- The use of a random-effects model is appropriate for accounting for inter-study variability. However, the authors mention significant heterogeneity in some outcomes but do not explore potential sources of heterogeneity in depth, such as geographic region or sport type (team vs. individual sports).

5- The findings—female athletes reporting higher levels of anxiety, depression, and distress—align with broader mental health trends, yet the causal interpretations are limited due to the cross-sectional nature of most studies. whats your justification?

6- The discussion effectively highlights the unique challenges faced by female athletes, including financial and career uncertainties. However, the article could have offered more concrete recommendations for sports organizations on how to address these issues, especially in light of the highlighted mental health disparities.

Reviewer #3: The authors investigated the sex differences of the impact of the C19 pandemic in elite athletes through a systematic review and meta-analysis. The results showed that female athletes reacted differently to the C19 lockdowns with higher levels of anxiety, depression and stress. However this didn’t have any influence on their alcohol use, nor eating disorders, gambling, substance use or sleep problems.

Clear and concise systematic review with meta-analysis.

L234. Maybe mention that alcohol use was different in one study (Imboden et al.) of this meta-analysis.

6. PLOS authors have the option to publish the peer review history of their article (what does this mean?). If published, this will include your full peer review and any attached files.

Reviewer #1: No

Reviewer #2: **Yes: **Morteza Taheri

Reviewer #3: No

---

## [Author Response · Author response to Decision Letter 0]

9 Oct 2024

We have provided a file with point by point responses.

---

## [Decision Letter · Decision Letter 1]

20 Nov 2024

Exploring Mental Health Symptoms in Elite Athletes During the COVID-19 Pandemic: A Systematic Review and Meta-Analysis on Sex Differences

PONE-D-24-28145R1

Dear Dr. Ling

We’re pleased to inform you that your manuscript has been judged scientifically suitable for publication and will be formally accepted for publication once it meets all outstanding technical requirements.

Kind regards,

Efrem Kentiba, PhD

Academic Editor

PLOS ONE

Additional Editor Comments (optional):

Reviewers' comments:

Reviewer's Responses to Questions

**Comments to the Author**

1. If the authors have adequately addressed your comments raised in a previous round of review and you feel that this manuscript is now acceptable for publication, you may indicate that here to bypass the “Comments to the Author” section, enter your conflict of interest statement in the “Confidential to Editor” section, and submit your "Accept" recommendation.

Reviewer #2: All comments have been addressed

Reviewer #3: All comments have been addressed

2. Is the manuscript technically sound, and do the data support the conclusions?

Reviewer #2: Yes

Reviewer #3: Yes

3. Has the statistical analysis been performed appropriately and rigorously? 

Reviewer #2: Yes

Reviewer #3: Yes

4. Have the authors made all data underlying the findings in their manuscript fully available?

Reviewer #2: Yes

Reviewer #3: Yes

5. Is the manuscript presented in an intelligible fashion and written in standard English?

Reviewer #2: Yes

Reviewer #3: Yes

6. Review Comments to the Author

Reviewer #2: The revisions are accepted. now your article been more completed and the necessary criteria as noted is good

Reviewer #3: (No Response)

7. PLOS authors have the option to publish the peer review history of their article (what does this mean?). If published, this will include your full peer review and any attached files.

Reviewer #2: **Yes: **morteza taheri

Reviewer #3: **Yes: **Samuel Iff

---

## [Editor Report · Acceptance letter]

1 Jan 2025

PONE-D-24-28145R1 

PLOS ONE

Dear Dr. Ling, 

I'm pleased to inform you that your manuscript has been deemed suitable for publication in PLOS ONE. Congratulations! Your manuscript is now being handed over to our production team.

Kind regards, 

on behalf of

Dr. Efrem Kentiba 

Academic Editor

PLOS ONE